Theoretical study of ArcB and its dimerization, interaction with anaerobic metabolites, and activation of ArcA

Padilla-Vaca Felipe 1
de la Mora Javier 2
http://orcid.org/0000-0001-8475-2282 García-Contreras Rodolfo 3
http://orcid.org/0000-0003-2780-5223 Ramírez-Prado Jorge Humberto 4
Vicente-Gómez Marcos 1
Vargas-Gasca Francisco 1
Anaya-Velázquez Fernando 1
Páramo-Pérez Itzel 1
Rangel-Serrano Ángeles 1
Cuéllar-Mata Patricia 1
Vargas-Maya Naurú Idalia 1 ni.vargas@ugto.mx
http://orcid.org/0000-0003-4332-3734 Franco Bernardo 1 bfranco@ugto.mx
1 Biology, Universidad de Guanajuato , Guanajuato, Guanajuato , México
2 Genética Molecular, Instituto de Fisiología Celular , Mexico City, Mexico City , México
3 Facultad de Médicina, Universidad Nacional Autónoma de México , Mexico City, Mexico City , Mexico
4 Unidad de Biotecnología, Centro de Investigación Científica de Yucatán, A. C. , Mérida, Yucatán , México
Gillespie Joseph
Electronic publication date: 2023 Oct 13
Publication date: 2023
Volume: 11
Electronic Location ID: e16309
Received 2023 May 17; Accepted 2023 Sep 27
Copyright: © 2023 Padilla-Vaca et al.
Copyright year: 2023
Copyright holder: Padilla-Vaca et al.
License: This is an open access article distributed under the terms of the Creative Commons Attribution License, which permits unrestricted use, distribution, reproduction and adaptation in any medium and for any purpose provided that it is properly attributed. For attribution, the original author(s), title, publication source (PeerJ) and either DOI or URL of the article must be cited.
License URL: https://creativecommons.org/licenses/by/4.0/

Keywords: ArcB, Two-component systems, Protein modeling, AlphaFold2, Signaling mechanism, Kinase regulation

Funding: The authors received no funding for this work.

==============================
The complex metabolism of Escherichia coli has been extensively studied, including its response to oxygen availability. The ArcA/B two-component system (TCS) is the key regulator for the transition between these two environmental conditions and has been thoroughly characterized using genetic and biochemical approaches. Still, to date, limited structural data is available. The breakthrough provided by AlphaFold2 in 2021 has brought a reliable tool to the scientific community for assessing the structural features of complex proteins. In this report, we analyzed the structural aspects of the ArcA/B TCS using AlphaFold2 models. The models are consistent with the experimentally determined structures of ArcB kinase. The predicted structure of the dimeric form of ArcB is consistent with the extensive genetic and biochemical data available regarding mechanistic signal perception and regulation. The predicted interaction of the dimeric form of ArcB with its cognate response regulator (ArcA) is also consistent with both the forward and reverse phosphotransfer mechanisms. The ArcB model was used to detect putative binding cavities to anaerobic metabolites, encouraging testing of these predictions experimentally. Finally, the highly accurate models of other ArcB homologs suggest that different experimental approaches are needed to determine signal perception in kinases lacking the PAS domain. Overall, ArcB is a kinase with features that need further testing, especially in determining its crystal structure under different conditions.

Introduction

Escherichia coli is a versatile organism that quickly adapts to changing environments, especially transitioning from aerobic to microaerophilic or anaerobic conditions. Its ability to use a variety of electron donors and acceptors is essential for its survival (Trotter et al., 2011). This metabolic flexibility makes E. coli a great candidate for producing biofuels and other chemicals through metabolic engineering, especially in low-oxygen environments (Yang et al., 2021; Ruiz et al., 2013). Therefore, understanding the regulation of E. coli metabolism is critical for engineering strains to produce valuable biotechnological applications.

Two-component systems (TCSs) are a widely conserved signaling mechanism in bacteria that allow them to sense and respond to environmental changes. Phosphorylation is the most prevalent protein modification (Papon & Stock, 2019). More than 300,000 TCSs have been identified in diverse organisms except for animals (Papon & Stock, 2019). TCSs consist of a sensor histidine kinase (HK) protein and a response regulator (RR) protein. The HK protein plays a crucial role in detecting changes in the environment, such as alterations in pH levels, osmolarity, or the presence of certain chemicals. Once the HK protein senses such changes and determines the appropriate response, it initiates a phosphotransferase reaction, resulting in the phosphorylation of the RR protein. Previous research has extensively studied this process (Stock, Robinson & Goudreau, 2000; Papon & Stock, 2019).

TCSs are involved in various physiological processes in bacteria, including chemotaxis, metabolism, and pathogenesis. Similarly, the PhoP-PhoQ system in Salmonella enterica is highly conserved in Gram-negative bacteria and plays a crucial role in virulence by regulating gene expression necessary for survival within host cells (Groisman, Duprey & Choi, 2021). This system can sense multiple signals relevant to the organism’s survival.

TCSs are also important targets for developing new antibiotics, as they are essential for the survival and growth of many pathogenic bacteria and are absent in animals. Inhibiting the activity of TCSs can disrupt key physiological processes, making it difficult for the bacteria to survive without affecting its animal host (Stephenson & Hoch, 2002; Dehbanipour & Ghalavand, 2022). For example, the case of daptomycin, an antibiotic that targets the signaling proteins of TCS in Gram-positive bacteria (Baltz, 2009).

The ArcA/B TCS comprises two proteins, the ArcB HK and the ArcA RR. This system was discovered by mutations that caused the increase of enzymes typically produced during aerobic growth when grown anaerobically in spontaneous mutants. This led to the discovery of ArcA (Iuchi & Lin, 1988) and ArcB (Iuchi, Cameron & Lin, 1989). This system plays a central role in sensing the redox state of the quinone pool (Georgellis, Kwon & Lin, 2001), contrary to the proposed role of the H+ gradient (Bogachev et al., 1995). The ArcA/B system activates or represses genes related to the transition from aerobic to anaerobic conditions and other genes involved in energy metabolism, transport, catabolism, and biofilm formation (Liu & De Wulf, 2004). The ArcB HK has important structural features that have been extensively studied, including a short periplasmic domain (Kwon, Georgellis & Lin, 2000) that is not typical in sensor kinases, a PAS domain, and a multistep phospho-relay system (Kwon et al., 2000; Yamamoto et al., 2005; Georgellis, Lynch & Lin, 1997; Teran-Melo et al., 2018). Observing the presence of a short periplasmic domain led to exploring the role of two cysteine residues in a putative PAS domain. One of these residues (specifically Cys180) has been identified as the key regulator mechanism of the kinase activity (Malpica et al., 2004).

The development of AlphaFold2, a predictive algorithm capable of providing accurate protein models, represents a significant advance in protein structure prediction (Jumper et al., 2021). This software uses an AI-powered algorithm that employs a neural network and Multiple Sequence Alignment statistics to predict the protein structure. It then uses an end-to-end solution to predict a protein folding covering the whole sequence (Marcu, Tăbîrcă & Tangney, 2022). The models generated by AlphaFold2 have several advantages over other protein structure modelers. For instance, AlphaFold2 provides better accuracy than template or homology-based modelers. Models compared to experimentally determined protein structures provide accurate models and can predict multimer complexes (Bertoline et al., 2023). The results published so far position AlphaFold2 as adequate for functional analysis. However, it should be noted that AlphaFold2 has some limitations. For example, it can have difficulty predicting the structure of disordered protein regions and loops, cannot predict novel structures or folding, and cannot identify defects in protein folding due to point mutations (Bertoline et al., 2023). Despite these limitations, AlphaFold2 is a powerful tool that can be used in combination with other methods to obtain more accurate predictions of protein structure and function.

The AlphaFold2 models in this study have provided valuable insights into the features of the ArcB sensor kinase. Protein models enable researchers to develop new hypotheses that can be confirmed by site-directed mutagenesis experiments, in this case, to further study ArcB in vitro and in vivo. By investigating the features of ArcB using AlphaFold2-generated models, further structural features that agree with biochemical and genetic data reported previously by other research groups. In the context of ArcB/A TCS, the AlphaFold2 models in this work revealed how the ArcB kinase dimer might be formed in vivo. Blind cavity detection (CB-Dock) analysis identified two putative regulatory cavities for binding anaerobic metabolites, which have been shown to regulate kinase activity. The data from these models have also been correlated with extensive biochemical and genetic evidence of the phosphotransfer reactions. Additionally, the structural features of different homologs of ArcB suggest that the signal detection mechanism in kinases lacking the PAS domain is more complex than previously thought and requires further experimental analysis.

Materials and Methods

The overall strategy to analyze the predicted structure of the ArcB/A TCS and the tools used for each task in this work are summarized in Fig. S1.

ArcA/B models

Protein models were either downloaded from the AlphaFold database hosted at EBI (https://alphafold.ebi.ac.uk/) or generated in the collaborative resource on line (https://colab.research.google.com/github/sokrypton/ColabFold/blob/main/AlphaFold2.ipynb) (Mirdita et al., 2022), without using template mode and concatenating the sequences by using “:” for each chain modeled as dimers (for ArcB or ArcA separately) or higher order oligomers (homodimers or heterooligomers using ArcA) (Evans et al., 2021). All the remaining settings were used in the default mode. GPU was used with extended RAM for oligomeric models. All models generated are provided as Supplemental Files (Supplemental Zip File). Chimera was used to measure the distance between Cys180 in the PAS domain and the two monomers in the dimer model. Model visualization was conducted with PyMol (Schrödinger & DeLano, 2020). Reference structures were obtained from the Protein Data Bank (https://www.rcsb.org/) as PDB files, and the PDB ID numbers are indicated on the figure legends.

ArcB docking with anaerobic metabolites

ArcB protein and ligand binding prediction was conducted with CB-Dock (Liu et al., 2020), a cavity-detection blind docking algorithm, that predicts the binding of ligands in a protein without knowing information about the binding sites and their properties. One advantage of this algorithm is that it predicts a curvature-based detection approach and performs docking with Autodock Vina achieving a ~70% success rate. Also, this algorithm performs a blind docking, providing a non-biased result. The CB-Dock server was used with the default settings. Docking data is presented in Table 1 showing the Vina score (the weighted sum of atomic interactions; McNutt et al., 2021) and cavity size (in Å). Images were obtained on the same web server. The structures of metabolites were downloaded as sdf files from PubChem and prepared for docking in UCSF Chimera as previously described (Butt et al., 2020; Pettersen et al., 2004) using the Dock Prep tool to add hydrogens and determine the charge and are included in the Supplemental File. The binding of metabolites with the isolated PAS domain was conducted. The PAS domain was modeled, including part of the connecting alpha helix region (residues 138–280 of the full-length protein) with AlphaFold2 as described in the previous section. Ligand physicochemical properties were calculated in MolCalc (Jensen & Kromann, 2013) by text mining the target molecule and using the default settings to calculate the properties of the ligands; images of dipole charge distribution were also generated in the same server and obtained as PNG files.

Table 1 Anaerobic ligand cavity features for full-length and PAS domain of the ArcB sensor kinase.

Anaerobic metabolite	Vina score	Cavity size (Å)	
Full-length ArcB	
D-lactate	−4.4	3,400	
L-Lactate	−4.5	3,400	
Butyrate	−4.3	3,400	
Pyruvate	−4.1	3,400	
Succinate	−4.6	3,400	
Formate	−2.9	3,400	
Ethanol	−2.6	3,400	
Quinone (Q0)	−6.1	3,400	
Ubiquinone	−9.8	3,400	
Menaquinone	−8.5	3,400	
PAS domain only	
D-lactate	−3.7	189	
L-Lactate	−3.7	708	
Butyrate	−3.9	708	
Pyruvate	−3.8	708	
Succinate	−4.2	708	
Formate	−2.2	708	
Ethanol	−2.4	708	
Quinone (Q0)	−5	708	
Ubiquinone	−7.5	64	
Menaquinone	−5.6	100	

Leucine zipper analysis

For assessing putative leucine zipper sequences, amino acid sequences of 30 HK of E. coli (obtained from KEGG; Kanehisa & Goto, 2000) were analyzed with GLAM2 (Frith et al., 2008) with the default settings for motif discovery using gapped alignments. The best score predicted motif (e value) for each HK was mapped to the whole sequence of each HK and identified in the protein sequences using MAST (Frith et al., 2008) with the default settings. The predicted motif was indicated also in the multimer model of ArcB by manually highlighting the residues in the protein using PyMol (Fig. S2).

ArcB homologs analysis

ArcB HK homologs analysis was conducted based on the report by Jung et al. (2008), where two types of ArcB HK have been identified. Protein structural models were downloaded from the AlphaFold database using the accession numbers reported by Jung et al. (2008) or modeled for those not available, for Type 1: Photobacterium profundum and Type 2: Mannheimia succiniciproducens ArcB homologs. Protein structure comparisons were carried out with mTM-align using the default options to highlight common core regions (Dong et al., 2018). US-align was also used with the default settings for accurate global alignments and to determine if the topology is shared among the two types of ArcB kinases (Zhang et al., 2022).

Protein models were visualized with PyMOL or Chimera (Schrödinger & DeLano, 2020; Pettersen et al., 2004).

Results

ArcB is one of the best-characterized HK in bacteria. The protein has been extensively studied and shown to have two transmembrane domains, a PAS domain and the multistep phosphorelay domains (Fig. 1A) (Iuchi et al., 1990; Georgellis, Lynch & Lin, 1997; Matsushika & Mizuno, 2000). The sensed signal identified is the redox state of the quinone pool (Georgellis, Kwon & Lin, 2001; Alvarez, Rodriguez & Georgellis, 2013). The actual sensing mechanism has also been elucidated. It involves two cysteine residues at the PAS domain (Malpica et al., 2004), where Cys180 is the most critical residue for forming a disulfide bond that is required for kinase inactivation under anaerobic growth conditions. Although some protein structures are available, they encompass only the phosphotransfer domain (Hpt) (Kato et al., 1999; Ikegami et al., 2001), and little is known about the overall folding of the ArcB kinase and the full-length ArcA response regulator (Toro-Roman, Mack & Stock, 2005).

Figure 1 ArcB kinase features and model analysis.

(A) Scan Prosite analysis of the coding sequence of ArcB is shown. Abbreviations: TM, transmembrane domains; LZ, Leucine zipper; PAS, Per-Arnt-Sim. The numbering on the top indicates the range spanning each element. The number below each domain indicates each section’s position on the full model in B. (B) ArcB model (AlphaFold database accession AF-P0AEC3-F1) in pLDDT color scheme. (C) The structural alignment of ArcB AlphaFold2 model and PDB 2KSD, the transmembrane structure of ArcB determined by NMR (TM-score 0.528, RMSD of 1.91 Å). In blue is PDB 2KSD, and in green is ArcB AlphaFold model. (D) Structural alignment using PDB structures of the Hpt domain, accession numbers 1A0B, 1FR0, 2A0B, and the full-length ArcB AlphaFold2 model. In magenta is shown the conserved regions (TM-score 0.940, RMSD, 0.87 Å). In blue, PDB 1A0B; in green, PDB 1FR0; in red, 2A0B and in purple, the ArcB AlphaFold2 model.

The full-length ArcB model

The ArcB kinase is comprised of a short cytoplasmic domain (residues 1 to 22), two transmembrane helices (TM) (residues 22 to 41 for TM helix 1 and 58 to 77 for TM helix 2), a short periplasmic domain between the two transmembrane domains (residues 42 to 57), and a PAS domain (residues 177–267), where the two cysteine residues are located (180 and 241). The catalytic domain is located between residues 290 to 520 (primary transmitter domain or H1), where the ATP binding box is located (spanning residues 399 to 473) and the primary transmitter residue (His292). The primary receiver domain (or D1) is located between residues 520 to 640, where the conserved Asp residue is situated (576). The phosphotransfer domain (Hpt) is in residues 640 to 778, with a conserved histidine residue (717) located within this region (Alvarez & Georgellis, 2010). The most relevant features are indicated in Fig. 1A.

To gain insight into the accuracy of the AlphaFold2 models, a series of comparisons were carried out to regions of the kinase that have been determined experimentally. The ArcB AlphaFold2 model showed that it is highly accurate in the full-length protein (Fig. 1B). In Fig. 1B, is shown in pLDDT color scheme. This score evaluates distance differences of all atoms in a model, including validation of stereochemical plausibility (Mariani et al., 2013), indicating expected distance error in Å. The less confident sections of the protein are the transmembrane domain and the histidine-containing phosphotransfer (Hpt) domain connecting helix with the PAD domain and the primary receiver domain.

The model was compared with the backbone structure of the membrane domain (PDB ID: 2KSD, Maslennikov et al., 2010), which NMR determined. As shown in Fig. 1C, the structural alignment of the PAS domain showed an RMSD of 1.91 but a TM-score of 0.528. The alignment may be explained by the loose helical packing of this kinase’s TM domain. In this case, a more flexible transmembrane domain 2 is found in the experimentally determined structure, perhaps due to the technique used (solid-state NMR, recorded at 45 °C) and the analysis method that lacked detergents, so the structure may be affected by hydrophobic hindrance. Additionally, the secondary transmitter domain or Hpt domain (Fig. 1D) was compared with the three available structures of this domain (PDB 1FR0, 2A0B, and 1A0B) (Kato et al., 1997, 1999; Ikegami et al., 2001). The comparison revealed an RMSD of 0.87 Å and a TM-score of 0.94, which suggests a nearly identical structure, even when including PDB 1FR0, which was the Hpt domain determined by NMR and shown to be a highly dynamic structure. The overall structural alignment with experimentally determined structures suggests that the prediction made by AlphaFold2 agrees with experimentally determined structures and is useful for further analysis.

Functional analysis of the ArcB model

The next structural feature to analyze is the leucine zipper, which was characterized previously by Matsushika & Mizuno (2000). For such analysis, the dimer model was generated using AlphaFold2. In Fig. 2A, the dimer is shown, each monomer in a different color. The leucine residues involved in the leucine zipper are indicated in Fig. 2B.

Figure 2 ArcB dimer model by AlphaFold2 multimer.

(A) The ArcB dimer model is shown; in magenta, one monomer, and in cyan, the other monomer, orientation is intended to show the position of all domains, as in Fig. 1. (B) Leucine zipper residues are indicated in both monomers in red. In the zoomed region, the cyan box indicates the residue that, when mutated, leads to the loss of regulation and, in magenta, the loss of kinase activity in vitro (Matsushika & Mizuno, 2000; Nuñez Oreza et al., 2012). Black arrows indicate the residues that mutations lead to a dominant negative phenotype in vivo. Leucine numbering is displayed on the right of the figure.

As shown previously, Nuñez Oreza et al. (2012) identified key residues in the helix spanning the cytoplasmic domain after the second transmembrane domain (residues 70–121) that are needed for the signal transduction mechanism. In Fig. 2B, the multimer model shows that all the leucine residues face each other. The model presented by Nuñez Oreza et al. (2012) suggests that specific leucine residues play a crucial role in maintaining the stability of the dimer and facilitating the signal detection and transduction mechanism. Experimental evidence supports this model, particularly regarding these key leucine residues. When these residues are mutated, either the regulation by the quinone redox state is eliminated (cyan box in Fig. 2B) or the activity is completely lost (magenta box in Fig. 2B) (Matsushika & Mizuno, 2000; Nuñez Oreza et al., 2012). Also, the other leucine residues that in vivo render a dominant negative phenotype are located near the second TM domain (Fig. 2B, black arrows). The mechanism that generates a dominant negative phenotype may be related to taking further apart the PAS domain from each monomer. The conformation leads to the in vitro and in vivo phenotype of the mutants in these residues or a null kinase activity as shown in vitro for the Leu87→Val mutant (Nuñez Oreza et al., 2012). The Leu102→Val mutant has higher activity in anaerobic conditions, suggesting that the kink found in the helix connecting the transmembrane domain with the PAS domain may acquire more flexibility, as seen in the multimer model (Fig. 2B). Also, the structural data are consistent with the predicted positions of the leucine residues proposed by Matsushika & Mizuno (2000), and the data with the report by Nuñez Oreza et al. (2012) is consistent with the regulatory effect of these leucine residues.

In Fig. S2A, the presence of leucine zippers found in 21 out of 30 sensor kinases from E. coli, indicates that ArcB leucine zipper is located spanning residues 20 to 41 (black arrow), which is in the second TM helix (Fig. S2B). The result suggests that the inner membrane leucine zipper may not be functional for establishing the kinase dimer, but the rest that has been experimentally validated is the real leucine zipper. Still, it may be related to stabilizing the TM domains only or interacting with specific lipids in the membrane.

Next, the PAS domain was analyzed for proper folding prediction compared to the structure of PAS domains determined experimentally. In Fig. 3A, the independent ArcB PAS domain was modeled with high accuracy spanning residues 139–279 (for numbering, refer to Fig. S3). In Fig. 3A, the structural comparison between the model (grey structure) and PAS domain alone (cyan) indicates that the two structures have identical TM-score 0.99119. The two cysteine residues needed for kinase regulation (Malpica et al., 2004) are located near the connecting helix between TM domain 2 and the rest of the kinase, Cys180 in the upper and Cys241 in the lower part of the PAS domain (Fig. 3B).

Figure 3 PAS domain architecture in ArcB analysis.

(A) The PAS domain, was modeled individually. Here, the rank 1 model of the PAS domain (residues 138–280) is shown. The cyan structure indicates that the individual model renders the same structure as in the model of the full-length sensor kinase, TM-score of 1. (B) Structural alignment with PAS domains from other sensor kinases (the conserved region in the first image is shown in magenta): PDB files indicated by color: 1DRM (FixL sensor kinase, PAS domain ligand-free), 1II8 (PAS kinase N-terminal domain, Homo sapiens), 1XJ3 (FixL sensor kinase PAS domain, unliganded ferrous form), 1P97 (HIF2a transcription factor, mammalian transcription factor involved in oxygen sensing), 2W0N (DcuS sensor kinase, E. coli), 3A0S (ThkA sensor kinase, Thermotoga maritima), 3CWF (extracytoplasmic PAS like domain in PhoR kinase, Bacillus subtilis), 5HWV (Tod’s sensor kinase, Pseudomonas putida,) and ArcB PAS domain model. Alignment metrics: TM-score 0.528; RMSD, 1.91 Å. (C) In gold, PAS domain and in cyan, Cys180, and Cys241 residues. (B) The position of the two cysteine residues needed for kinase regulation is indicated (in red). (D) The comparison of Cys180 residue in a Asn181 to Ala mutant (Matsushika & Mizuno, 2000). (D) Color scheme: Cyan, Wt kinase. Grey, Asn181®Ala mutant PAS domain. Red, Cys180 in Wt kinase. Green, Cys180 in mutant PAS. Purple, Asn181 in Wt PAS domain. Yellow, Ala residue in mutant PAS domain. TM-score: 0.99591.

Further, to assess the structural conservation of the PAS domain, the 139–279 model was aligned with eight PAS domains whose structures had been experimentally determined. As shown in Fig. 3C, the PAS domain of ArcB exhibits a conserved core domain with other PAS domains from bacteria and eukaryotic organisms, suggesting that this domain may also be involved in the fine-tuning of the activation or inhibition of the kinase (see below).

Previously, Matsushika & Mizuno (2000) reported that Asn181→ Ala mutant showed a similar phenotype to the one displayed by an ArcB lacking the complete PAS domain (ΔPAS). In Fig. 3D, we show that the PAS domain mutant and wild type in this residue are identical in folding, and the location of the Cys180 residue is identical in both predicted structures. The resulting null phenotype found by Matsushika & Mizuno (2000) is unclear by both the biochemical and genetic data as well as for the predicted structural folding of the PAS domain bearing this mutation and the resulting position of the cysteine residues.

The activation and silencing of the kinase depend on two cysteine residues (180 and 241, Malpica et al., 2004). To date, no structural data has been provided to link the disulfide bridge formation between Cys180 pairs in the kinase dimer, except functional analysis using a hybrid kinase with the membrane domain of the Tar chemoreceptor showing that the kinase has a rotational on/off mechanism (Kwon, Georgellis & Lin, 2003). The AlphaFold2 multimer shows that Cys180 residues are oriented apart in the predicted model (Fig. 4A), and the linear distance between the two residues is over 25 Å. Biochemical studies have estimated that the optimal distance between two cysteine residues is in the range of 2.4 and 4 Å (Bhattacharyya, Pal & Chakrabarti, 2004) and is dependent on the environment surrounding the cysteine residue (Bhattacharyya, Pal & Chakrabarti, 2004). As shown in Fig. 4B, the kinase shows a 25.811 Å in a linear distance. However, as demonstrated by Kwon, Georgellis & Lin (2003), the rotation of ArcB can be up to 100°. Thus, these two residues are likely to be in proximity to form a disulfide bond.

Figure 4 ArcB dimer analysis shows that the predicted arrangement is consistent with the active form of the kinase.

(A) The position of the cysteine residues on each subunit using the ArcB dimer represented in cartoon mode. In the close-up, the linear distance measurements between Cys180 were determined in USCF Chimera. In cyan, Cys180 is shown, and in red, Cys241. (B) The position of the first cytoplasmic residues and the cysteine residues involved in the kinase activation in the dimer model, which is in agreement with the position of the active form of the kinase (Kwon, Georgellis & Lin, 2003). The residues are indicated by position, and the model was stripped of the rest of the regions, Cys180 is indicated in red. (C) The tilted upper view of Cys180 in cyan and two aromatic residues, Tyr170 and Phe177, are shown in magenta and orange, respectively.

To gain further insight into whether the model is positioned in an active or inactive state. In the AlphaFold2 dimer model, the first residues of the cytosolic domain are stripped of the rest of the protein, analyzed from the cell membrane point of view, and tilted towards the front to have, in perspective, the Cys180 residues (Fig. 4B). The leucine residues are positioned as Kwon, Georgellis & Lin (2003) suggested in the regulable form of the kinase (residues Gln79, Leu80, and Glu81), with the two cysteine residues pointing apart. The structural model suggests that at least the rotation must bring up to 2–4 Å apart each monomer to form the disulfide bond between the two Cys180. Once assembled, this will bring the two catalytic domains apart and stimulate the phosphatase activity (which has been previously shown in Malpica et al. (2004)), and the model is in the off state but orientated in the regulatable form of the kinase (Kwon, Georgellis & Lin, 2003). Therefore, the model is consistent with a rotational on/off mechanism, and this is required for the disulfide bond formation between Cys180 on each monomer and a 100° rotation to render the kinase active. Cys241 is on the same surface and partially involved in the signal-sensing mechanism (Fig. 4A). Figure 4B shows that Cys241 is buried deeper in the PAS domain, explaining its lesser role in kinase inactivation.

Figure 4C shows that the Cys180 residue is located near Tyr170 and Phe177. Aromatic residues are relevant for the disulfide bond formation since the presence of an aromatic ring favors its formation (Bhattacharyya, Pal & Chakrabarti, 2004). In Fig. 4C, the Cys180 (cyan) and Tyr170 (magenta) are highlighted in the dimeric structure of ArcB, indicating that the Tyr170 residue delimits the orientation of the disulfide bond formation and the stability since its located towards the Cys180 residue on each ArcB monomer. Phe177 may have a lesser role since its orientation is farther apart from the cysteine residue (Bhattacharyya, Pal & Chakrabarti, 2004).

The binding of anaerobic metabolites and the relevance of D-lactate

Docking methods allow the detection of putative binding sites of molecules relevant to the function of a protein or to screen for drug discovery (Tripathi & Bankaitis, 2017). CB-Dock, an algorithm that predicts the binding of ligands in a protein without knowing the binding sites’ properties, was used to detect putative cavities where anaerobic metabolites might bind (Liu et al., 2020). The approach used for this analysis was using this method with quinone and menaquinone molecules as recognized regulators of the kinase activity in the full-length kinase or the ArcB PAS domain (Georgellis, Kwon & Lin, 2001; Liu et al., 2020; Cao & Li, 2014) and then evaluate other anaerobic metabolites for binding. This approach will allow us to analyze whether the PAS domain, with a highly conserved structure (Fig. 3C) or in the full-length kinase, there are high probability binding sites for anaerobic metabolites. Also, the exact binding of D-lactate, for instance, has not been determined. The catalytic domain may be active for binding other molecules or perhaps an allosteric pocket that regulates kinase ATP binding or phosphotransfer reaction efficiency present. We began by determining the binding of Q0, which has been frequently utilized in vitro assays and causes reduced hindrance due to its lack of a long aliphatic chain. Figure 5A indicates that Q0 and menaquinone can bind in the catalytic domain, while Fig. 6A shows that it can also bind in the PAS domain.

Figure 5 D and L-lactate occupancy in a putative cavity in the full-length ArcB kinase catalytic domain.

(A) The full-length ArcB model was used for CB-Dock analysis using Q0 or Ubiquinone, both exerting a well-characterized regulatory effect on the kinase activity. The position of each ligand is indicated. (B) CB-Dock analysis of D or L-lactate conformers binding to the full-length ArcB kinase.

Figure 6 D and L-lactate occupancy in a putative cavity in the ArcB PAS domain.

To assess the binding of metabolites in the PAS, a minimal version of this region was modeled by AlphaFold2 (residues 140–280). (A) The binding of Q0 is shown, a close-up is included to highlight the position of the aromatic ring. Red asterisk indicates the position of Cys180. (B) Shows the binding of ubiquinone (2-octa prenyl-3-methyl-5-hydroxy-6-methoxy-1,4-benzoquinone). (C) Putative binding cavities for D- and L-lactate, respectively. The ligands are colored grey and red.

Second, in the case of ubiquinone, 2-octa prenyl-3-methyl-5-hydroxy-6-methoxy-1,4-benzoquinone was used and is the corresponding ubiquinone synthesized in E. coli (Kwon, Kotsakis & Meganathan, 2000). This molecule can bind to the catalytic and PAS domains according to CB-docking (Figs. 5A and 6B). Both Q0 and ubiquinone bind to a similar pocket in the PAS domain, with the aromatic ring directed towards the Cys180 location (indicated by the red asterisk in Fig. 6). The same observation was found for menaquinone (Fig. S4 for the full-length protein and Fig. S5 for the PAS domain). According to the biochemical data, the binding domain is the PAS domain (Malpica et al., 2004). However, the CB-Dock tool data indicates ubiquinone is the most probable binding molecule (Table 1) in the full-length and PAS domain.

With the above data, then the binding of anaerobic metabolites was explored. Caution must be taken since the binding of quinone molecules was predicted to bind in the catalytic and PAS domains. In the case of anaerobic metabolites, one possibility is that the metabolite D-lactate is sensed in the catalytic domain since it has been shown to enhance the kinase activity both in vitro and in vivo (Georgellis, Kwon & Lin, 1999; Rodriguez, Kwon & Georgellis, 2004). CB-Dock tool prediction with the full kinase structure, D-lactate is predicted to bind in the deep pocket found in the ATP-binding domain (residues in the N, G1, and G2 boxes) (Fig. 5B). However, this is also predicted in the isolated PAS domain (Fig. 6C, for the residues involved in the binding, refer to Fig. S3). In the PAS domain case, the prediction suggests that binding is in a cavity with charged residues. Thus far, no experimental evidence of the binding sites of D-lactate has been provided. These two cavities in the sensor kinase indicate the possibility of utilizing two mutagenesis strategies to investigate the regulation mechanism. We suggest the following approaches: (1) conducting site-directed mutagenesis of the residues in the ATP binding boxes that do not affect the in vitro phosphorylation rate, and (2) performing mutagenesis targeting the 78–778 region of the kinase that demonstrates quinone-mediated regulation in vitro. Directing mutations to the PAS domain may uncover further evidence favoring or discarding this analysis and ultimately showing the regulation exerted by these anaerobic metabolites.

When the same analysis is conducted with L-lactate and other anaerobic metabolites (Figs. 5, 6, S4 and S5), the kinase can bind in the same predicted regions as D-lactate. L-lactate is capable of binding in both the ATP-binding region and the PAS domain (Figs. 5 and 6). Table 1 shows the differences in the binding score and cavity size, where other anaerobic metabolites (butyrate, pyruvate, succinate, formate, and ethanol) are bound either in the ATP-binding region or in the PAS domain (Figs. S5 and 6, respectively). The results show that D-lactate has a high probability cavity, and binding for D-lactate and L-lactate can also bind to the same pocket in the same region but form a less strongly bound due to a reduced number of contacts. Butyrate and pyruvate also show a probability like D-lactate (Table 1). However, in the PAS domain, the results are inverted; L-lactate has an equal binding probability and a bigger cavity to bind. The results from CB-Dock suggest that the two regions may regulate kinase differently; an interesting experimental approach could be to analyze the autophosphorylation rate of mutants lacking the residues involved in binding either metabolite and the effect of oxidized and reduced quinone.

As shown in Table 1, the same cavity has a high probability of binding in the full-length kinase. Butyrate, pyruvate, and succinate occupy the same cavity size. Formate and ethanol showed a low likelihood of binding. These results are consistent with the PAS domain alone (Table 1). Overall, exploring the ATP binding box and the PAS domain involved in binding anaerobic metabolism-derived metabolites can further provide evidence of other regulatory conditions for the ArcB sensor kinase that are worth exploring experimentally.

The calculation of the overall characteristics of each metabolite is a useful approach to determining why the binding is probable in the molecules tested. In Table 2, the properties of these molecules are shown, and in Fig. S6, a graphical representation of the predicted charge distribution for each anaerobic metabolite tested is shown.

Table 2 Estimated physicochemical properties of anaerobic metabolites tested for binding to the ArcB sensor kinase.

Metabolite	Total solvation energy (kJmol−1)	Surface area (Å2)	Dipole moment	Total entropy (J mol−1 K−1)	
D-lactate	−24.60	263.8	3.72	321	
L-Lactate	−24.35	254.43	5.64	344.41	
Butyrate	−20.63	283.48	5.18	355.64	
Pyruvate	−16.69	240.57	2.84	340.19	
Succinate	−36.86	294.69	5.02	400.30	
Formate	−41.51	188.24	4.83	251.53	
Ethanol	−1.72	209.32	1.69	274.97	

The reduced binding of L-lactate can be explained by a stronger dipole (Table 2 and Fig. S6), which explains the position in the full-length kinase as bound outward from the identified cavity (Fig. 5), in an inverse manner to the PAS domain (Fig. 6). Despite the results presented here, a question remains regarding whether L-lactate can modulate the disulfide bond formation in Cys180 allosterically, like the experimental evidence shown for D-lactate (Georgellis, Kwon & Lin, 1999; Rodriguez, Kwon & Georgellis, 2004). The available evidence has not provided proof of L-lactate binding in vitro or in vivo due to the higher affinity for D-lactate. Also, the cavity size and Vina score for succinate and pyruvate is similar to D and L-lactate binding sites. However, both metabolites show a stronger dipole moment and higher total entropy (Table 2), suggesting that the binding of these two relevant anaerobic metabolites may depend on the cytoplasmic pH; perhaps the determination of ArcB activity under acidic conditions may render regulation by any of the tested anaerobic metabolites.

The properties of the rest of the anaerobic metabolites are similar, except for ethanol, which has the lowest dipole and solvation energy. The net charge is distributed along the molecule similarly to D-lactate, with the varying size of each molecule (Fig. S6). From a physicochemical point of view, the rest of the anaerobic metabolites are candidates for regulating ArcB at either the ATP binding box or PAS domain due to their similar chemical features. Putative binding cavities of the same size may, in vivo, provide further regulation of the kinase activity or sensitivity to the quinone pool redox state.

The interaction between ArcB and ArcA

ArcB exerts its action by activating ArcA via a phosphotransfer reaction with the conserved Asp54. To date, the only available structure of ArcB with a bound response regulator is to CheY (Yaku et al., 1997; Kato, Mizuno & Hakoshima, 1998). The reported interaction, however, is not between the cognate response regulator of ArcB. In Fig. 7, using the multimer capability of AlphaFold2, the predicted interaction of ArcA and ArcB is shown. In Fig. S7, the ArcA dimer model shows that the conformation predicted is consistent with the available crystal structures (ArcA receiver domain without and with beryllium fluoride to mimic phosphorylation, Toro-Roman, Mack & Stock, 2005). Also, the quality assessment of the five models of the ArcA tetramer is shown in Fig. S8. Unambiguously, biochemical studies have demonstrated that the autophosphorylation and then phosphotransfer route is from His292 to Asp576 of the same monomer of ArcB. The phosphotransfer from Asp576 to His717 occurs in an intermolecular reaction between the two monomers of ArcB and then the transfer to ArcA (Teran-Melo et al., 2018).

Figure 7 Multimer model of the dimeric ArcB kinase in interaction with ArcA.

(A) The dimer and tetramer models of ArcA, receiver Asp54 residue is indicated in bright green. (B) The full dimer ArcB is shown in cyan, and in orange, ArcA monomers. The white arrows indicate the domain in ArcA corresponding to the receiver domain. (C) Two variations in the prediction of ArcB in complex with two ArcA dimers. Each ArcA dimer pair is shown in orange and red-orange. (D) Two variations in the prediction of the ArcB (in cyan) dimer in association with ArcA tetramers. Each ArcA subunit is shown in a different color. (C and D) The ArcA monomers are arranged in the orientation of the Hpt domain in ArcB.

However, no structural data are available to determine in vitro and in vivo that the geometry and spatial organization of ArcB in the interaction with ArcA can ultimately resolve the phosphotransfer pathway, even though the extensive biochemical and genetic data support the reaction pathway (Kwon, Georgellis & Lin, 2000; Peña-Sandoval & Georgellis, 2010; Teran-Melo et al., 2018).

In order to predict the binding of ArcA to ArcB, the model of full-length ArcA was generated to assess its accuracy against the receiver domain of ArcA that has been experimentally determined (Toro-Roman, Mack & Stock, 2005). In Fig. 7A, the two reported conformations of ArcA were modeled (for model data, refer to Fig. S7A through C) as a dimer and a tetramer (Toro-Roman, Mack & Stock, 2005). The dimer model renders a near identical conformation with the crystal structure of the ArcA receiver domain (Figs. 7A and S7D, TM-score 0.96829). The tetrameric form of ArcA is found by size exclusion chromatography (Toro-Roman, Mack & Stock, 2005). The model also predicts a spatial orientation of each monomer as in the dimeric ArcA (Fig. 7A, model statistics in Fig. S8). The modeled ArcA is consistent with the structural information available (Toro-Roman, Mack & Stock, 2005).

Then, the ArcB interaction with ArcA was modeled. First, the kinase dimer was modeled with a monomeric form of ArcA bound to each subunit of ArcB (Fig. 7B, statistics of the model in Fig. S9). In this model, the receiver domain of ArcA is positioned in the Hpt domain of ArcB on each subunit, which is consistent with the vast experimental evidence on the phosphotransfer mechanism (Teran-Melo et al., 2018).

However, the DNA-binding form of ArcA is either a dimer or a tetramer (Toro-Roman, Mack & Stock, 2005). Therefore, modeling ArcB with either the dimeric (statistics in Fig. S10) or tetrameric (statistics in Fig. S11) form of ArcA may provide insight into the regulation of the response regulator activation.

Firstly, in the model contemplating an ArcA dimer, all the predictions indicate two possible interactions between ArcA and ArcB (Fig. 7C). The ArcA interaction is with the Hpt domain (one monomer) in all cases. The second molecule (Fig. 7B) demonstrates direct binding to ArcA in the catalytic domain. Two conformations of the ArcA monomers are predicted and shown in two shades of orange, with model statistics presented in Fig. S10. The models suggest that ArcA may need to be phosphorylated before dimerization, consistent with previous research (Toro-Roman, Mack & Stock, 2005).

Secondly, Toro-Roman, Mack & Stock (2005) found that upon phosphorylation, ArcA exhibits a higher-order structure as a tetramer. As shown here, the tetrameric ArcA can arrange in the same orientation as in the dimer crystal. This is consistent with the experimental determination of the activation domain and the activated form as a tetramer reported by Toro-Roman, Mack & Stock (2005). In Fig. 7, the interaction between the ArcB dimer and the monomer of ArcA is predicted to bind to the Hpt domain (Fig. 7B).

When the kinase and response regulator were modeled in different stoichiometries, i.e., as a dimer with one ArcA, with the ArcA dimer, and with the ArcA tetramer, the resulting models showed different architectures. Figure 7D shows the model of ArcB with the tetrameric form of ArcA, revealing two spatial distributions. The first is two dimers bound between the Hpt domain and the PAS domain (left image in Fig. 7D), while the second is a dimer bound to the Hpt and another dimer bound to the catalytic domain (right image in Fig. 7D). Taking together the data in Figs. 7C and 7D, the most plausible mechanism involves the phosphorylation of an ArcA monomer. Then it forms higher order complexes, perhaps as shown in Fig. 7C, one ArcA monomer is bound to the kinase awaiting phosphorylation. Further biochemical data is needed to elucidate a stepwise mechanism of ArcA activation.

The current model for phosphotransfer in ArcB is as follows: from H1 to D1, the phosphotransfer occurs in the same subunit, and then, the phosphate group is transferred to the Hpt domain (H2). Using the model presented in Fig. 8A, a specific focus was taken to assess the orientation of the catalytic residues in ArcB in relation to the position of ArcA. The location of key residues and the domains involved in each step of the phosphotransfer mechanism is presented in Fig. 8B. This model suggests that the active kinase interacts with the response regulator (ArcA, in green) to the H2 domain (in blue). In this model, H1 (orange) and D1 (in yellow) are placed in the same region of the dimeric ArcB, whereas the H2 domain is placed afar from D1. The predicted flexible part shown in the model explains the in vitro and in vivo demonstration that the final phosphotransfer pathway can occur by displacement of the Hpt domain (H2) towards D1. It is more flexible in the 290–778 kinase fragment (Fig. S12). In Fig. 1C, the ArcB dimer is a snapshot of an immobile ArcB.

Figure 8 Dimeric ArcB, in association with a monomer of ArcA, is consistent with the phosphotransfer mechanism.

(A) The position of the catalytic domains of ArcB: in yellow, the primary transmitter domain (H1); in orange, the primary receiver domain (D1); in blue, the secondary transmitter domain or Hpt; and in green, ArcA. The amplified image indicates the position of relevant residues of each domain, and on the right, the position of Asp54 in ArcA relative to His717 in the Hpt domain. Also, the residues in the H1 and D1 domains are indicated, and the Cys180 residues on each ArcB monomer. (B) A close-up image of the orientation and closeness of ArcB HPt His717 residue with the receiver Asp54 in ArcA. (C) In the dimer model, the position of the two Hpt domains is located in the opposite orientation relative to Asp576, consistent with the phosphotransfer mechanism. In this model of the kinase showing the position of the two Hpt domains (in blue) suggests that the structure is consistent with the biochemical data showing that the forward phosphorylation reaction occurs from His292 to Asp576 (purple arrow over Asp576) of the same monomer of ArcB. Then, the phosphate is transferred to His717 in the Hpt domain (orange arrow) of the opposite monomer and transferred to Asp54 in ArcA since the Hpt flexible region faces the dimer’s opposite side (bright green arrow).

The crystal structure of the Hpt domain contains just the first part of the flexible region (Fig. 1). Therefore, the most significant limitation of protein models and crystals is that these regions’ dynamics are lost. That may also explain why no crystal structures spanning the complete cytosolic part of ArcB is available yet. The model presented here suggests that all the catalytic domains are highly mobile. For the Hpt domain, depicted in Figs. 8B and 8C show the side where the Asp576 residue resides and meets the opposite on each monomer. The Hpt domain can move towards this residue of the opposing subunit due to a flexible connector, consistent with the biochemical findings of Teran-Melo et al. (2018). One important feature of the models shown here is the position of the cysteine residues in the PAS domain; all models predict both Cys180 facing in the opposite direction, consistent with the active form of the kinase. One remaining question is whether ArcA, once bound, the phosphotransfer reaction occurs directly and the kinase is already locked in the conformation needed to be in close proximity to ArcA (as depicted in Fig. 8C) or, first, the Hpt domain is phosphorylated and this induces the displacemnt into the position shown in the model of Fig. 8A, allowing to bind ArcA and then transferring the phosphate group to ArcA Asp54 residue. In Fig. S13C, the monomeric form of ArcA with the predicted binding of ArcA shows that the Asp54 residue is closer to His717 than the position shown in the dimeric form of ArcB (Fig. 8B), showing the previously different phosphotransfer rate described previously between the signal decay using in vitro ArcB fragments of H1-D1-H2 (slower signal decay) vs. D1-H2 (faster signal decay) (Georgellis et al., 1998).

The reverse phosphotransfer in ArcB is relevant for signal decay (Georgellis et al., 1998) via the transfer from Asp54 of ArcA to the conserved histidine (His717) in the Hpt domain and then releasing the phosphate group from D1 domain (Georgellis et al., 1998; Peña-Sandoval, Kwon & Georgellis, 2005; Teran-Melo et al., 2018). In Fig. S13, the model of the monomer of ArcB interacting with ArcA shows a more flexed connector between the second transmembrane domain and the PAS domain (Fig. S13B), and the catalytic domains are positioned outwards (Fig. S13B). One relevant feature in this model is that the more open state is consistent with the reverse phosphotransfer seen in signal decay since the phosphor group travels in the revers phosphor relay reaction from D2 to H2 and then in D1 is released to the cytoplasm. This model shows that the position of Cys180 has shifted away from the second transmembrane domain and is now located opposite to its position in the dimeric kinase, as depicted in Fig. 4B. To confirm this, a model of residues 290–778 was generated, both alone and in complex with ArcA, as shown in Fig. S12. This model shows that the Hpt domain is less compact than in the dimer or the dimer in the complex with ArcA (Fig. 7). In this model, Asp576 is in the opposite position of His717, suggesting that a rotation in the same plane can occur in both forward and reverse phosphotransfer.

With the models presented here, the extensive biochemical and genetic data are consistent with the predicted structure of ArcB. The next step would be to determine the crystal structure in active and inactive states to assess the same transitional states of the Hpt domain in the active or inactive states of the kinase.

Structural features of ArcB homologs

The available protein models in the AlphaFold database and the comparison between the types of ArcB kinases can uncover mechanistic insights into signal sensing. Figure 9 shows a structural comparison between type 1 and 2 ArcB kinases (Jung et al., 2008). The distinction between the two types of ArcB kinase is proposed by Jung et al. (2008) by the lack of almost the entire linker region (residues 93 to 271, E. coli ArcB numbering, Uniprot P0AEC3), which includes the PAS domain. As proposed by Jung et al. (2008), the organisms bearing an ArcB without a PAS domain also lack the presence of ubiquinone, except for Pasteurella (Uniprot A0A126QD99), which is worth studying further to assess the mechanism of signal sensing.

Figure 9 Structural comparison of ArcB models from type 1 and 2 kinases.

(A) A comparison was conducted with US-aling. Type 1 kinases: in cyan, E. coli; in purple, Photobacterium profundum; in magenta, Yersinia pestis; in cornflower blue, Salmonella enterica; forest green, Photorhabdus luminescence; orange, Erwinia carotovora; dark grey, Shigella boydii. TM-alignment 0.645, RMSD 6.17 Å. Type 2 kinases comparison including ArcB from E. coli (Type 2 + ArcB legend): Cyan, E. coli; red, Mannheimia succinicproducens; green, Haemophilus influenzae; orange, Pasturella multocida; magenta, Actinobacillus succinogenes. TM-score, 0.4789, RMSD 5.83 Å. The differences between kinases are related to both structural and sequence features. Type 2 only, in purple, Mannheimia succinicproducens; blue, Haemophilus influenzae; orange, Pasturella mutocida; pink, Acitenobacillus succinogenes. TM-score 0.511, RMSD, 6.05 Å. (B) Type 2 ArcB homologs alignment; indicated in magenta, the position of cysteine residues in all kinases, arrow indicates the rotation on the same axis to show the back from the catalytic domains.

Regarding the possible sensing mechanism, the cysteine residue involved in the activation and deactivation of ArcB in E. coli is absent in type 2 kinases. Genetic data indicate that both Mannheimia (Uniprot A0A378NAL7) and Haemophilus (Uniprot A0A2S9SIC6) ArcB homologs can provide toluidine blue resistance and redox regulation in vivo in an arcB null mutant in E. coli respectively (Jung et al., 2008; Georgellis et al., 2001). However, Mannheimia ArcB in vitro fails to respond to ubiquinone and menadione or the regulation exerted by anaerobic metabolites (Jung et al., 2008). This suggests the need to explore further the residues identified in the CB-Docking analysis shown here in the PAS domain.

ArcB proteins from different bacterial species have variations in their amino acid sequences, including the number and location of cysteine residues. For example, the ArcB protein from Mannheimia has three cysteine residues located in the second transmembrane domain (Cys37), the linker region of D1 (Cys470), and the Hpt domain (Cys554). On the other hand, the ArcB protein from Haemophilus influenzae has five cysteine residues located in different domains of the protein. Specifically, there is one in the first transmembrane domain (Cys37, as in Mannheimia ArcB), one in the ATP-binding box and H1 domain (Cys268), one in the D1 domain (Cys472), and two in the Hpt domain (Cys574 and Cys596). Figure 9B shows that cysteine residues in type 2 ArcB proteins are clustered in either the second transmembrane domain or the D1 and Hpt domains (in magenta). Further investigation is required to determine the role of these cysteine residues and how they contribute to the signal transduction mechanism, possibly through cysteine disulfide bonding in the second transmembrane domain or interaction with other proteins that have yet to be identified.

Discussion

The orchestrated activation and repression of genes result in the adaptation of E. coli to different environmental cues. Despite all the knowledge gained from this model organism, several aspects of its physiology and molecular response to the environment need to be further scrutinized. One example of regulatory systems that still encase aspects that need further study is TCSs, such as their evolution, structure, specialization, and signaling mechanisms. In this report, we provide insight into the structural features of ArcB sensor kinase that confirms being studied in detail by genetic and biochemical approaches in numerous studies. Also, here we present evidence that predictive models can help advance our knowledge of the role of kinase regions involved in its regulation waiting for experimental structure determination. The predictive models also open the avenue for protein engineering applications such as developing biosensors (Alvarez & Georgellis, 2022; Rutter et al., 2021; Wang, Zhang & Childers, 2021) or even for biomedical applications (Woo et al., 2020).

The value of protein modeling

The models presented by AlphaFold2 and AlphaFold2 multimer consistently predict accurate structures that provide meaningful data or confirm robust experimental evidence, for which the only missing aspect is structural confirmation. Based on the available structures for ArcB, we confirmed that the structure contains highly flexible regions, the symmetry and architecture of the dimer are consistent with previous observations of the regulatory mechanism, and the interaction with ArcA is also compatible with genetic and biochemical data regarding the phosphorelay mechanism. The search for binding sites for anaerobic metabolites also uncovered putative target sites that have not been explored, specifically the ATP-binding box domain. This result is also consistent with type 2 kinases lacking the PAS domain, which fails to be regulated by D-lactate but can respond to the effect of external electron acceptors like ArcB from E. coli.

The short periplasmic domain in ArcB remains an important topic. Our analysis shows that the architecture of the membrane-spanning domains is tightly packed, with a kink in the second transmembrane domain (Fig. 1), which is relevant to forming the dimer and is stabilized by the leucine zipper (Fig. 2). The kinase activation may require additional aids for the rotation and avoiding diffusion in the membrane, which cannot be predicted from the generated model. We propose that the regulation of the rotation of the kinase may be related to the cell membrane structure, such as the effect of differential lipid concentration, type, and properties, including lipid raft-like structures. Guzmán-Flores et al. (2019) demonstrated the existence of detergent-resistant membranes in E. coli, but they did not find ArcB in these membrane domains. However, they did find ten proteins with unknown functions that may be associated with these domains and interact (perhaps weakly) with ArcB and other kinases, limiting their diffusion in the membrane plane. The same detergent-resistant domains have been shown to directly affect the kinase activity, as reported for the Staphylococcus aureus SaeRS TCS (Yeo et al., 2023). To test the effect of lipid raft-like structures on kinase activity, experiments using membrane-destabilizing compounds that may alter either the location or the functional dimers formed for signal sensing may shed light on the fundamental role of membrane micro-domains (García-Fernández et al., 2017).

The PAS domain is an ancient protein module found in all kingdoms of life (Nambu et al., 1991; Vogt & Schippers, 2015). Overall, the PAS domain structure is highly conserved among signal transduction proteins. In Fig. 3, we show the structural alignment of the ArcB sensor kinase with other PAS-containing proteins, both prokaryotic and eukaryotic, deposited in the PDB. The structure contains lower sequence conservation but high structural conservation, suggesting that the 3D space of this domain is relevant for its sensory role. Still, the specificity is constrained to the key residues involved in ligand binding. Also, this suggests that the sensory function of the PAS domain is broader than other more divergent signal sensory domains since, here, the regulatory Cys180 is a key element for signal perception. Also, exploring the Asn181→ Ala mutant in the PAS domain with the provided structural data suggests no major structural changes are observed. Perhaps the local physicochemical environment is needed for the proper oxidoreduction of Cys180. However, AlphaFold2 models must be taken with caution, specifically for the impact on ΔΔG values, since Pak et al. (2023) found that the impact on pLDDT values has low to no correlation with experimental ΔΔG values. Since no structural variations were found, Asn181 mutants with other residues are suggested to explore the impact of the physicochemical environment.

The effect of D-lactate has been addressed in vivo (Rodriguez, Kwon & Georgellis, 2004) and in vitro (Georgellis, Kwon & Lin, 1999). In both instances, the effect of D-lactate was observed in the full-length protein; thus, the exact pocket where the anaerobic metabolite exerts its influence remains unknown. Here, two putative pockets were identified, one in the catalytic H1 domain and the second associated with the PAS domain with lower probability. The full-length anaerobic metabolites were identified by cavity detection and blind docking to bind to the catalytic domain (Fig. 5). The model of the PAS domain alone reveals a potential cavity that could bind anaerobic metabolites and perhaps disturb the redox mechanism sensing, thereby enhancing kinase activity as previously shown (Malpica et al., 2004; Rodriguez, Kwon & Georgellis, 2004). Site-directed mutagenesis and biochemical analysis can be used to identify key residues for D-lactate binding and other anaerobic metabolites, which may be hidden from the high effect exerted by D-lactate. Ultimately, this can determine the main site for D-lactate binding and additional potential binding pockets both in vitro and in vivo.

Evolution of TCSs with emphasis on protein-protein contacts

One key question in the evolution of TCSs is their specificity and reduced crosstalk. Typically, a new TCS arises through operon duplication since the sensor kinase and response regulator are usually found in the same operon. This raises the question of how specificity is established, which can occur in two ways. The first is the evolution of a unique protein-protein interface that develops a new functional pathway. The second is through the independent evolution of two TCSs (the four proteins) to generate specificity (Nocedal & Laub, 2022), thus establishing a delimited regulatory pathway. In their work, Nocedal & Laub (2022) showed that three mutations are needed to develop the independence of a TCS.

Interestingly, these mutations occurred on the histidine kinase of one system and on the response regulator of the other, resulting in the evolution of specificity rather than changes in phosphotransfer speed between them. Based on the data presented in Figs. 7 and 8, it is less plausible that in vivo CheY can interact with the ArcB kinase (as shown in Yaku et al., 1997 and Kato, Mizuno & Hakoshima, 1998). The exerted effect on swimming behavior is related to the expression of flagellar genes (Kato et al., 2007) rather than a physical interaction between the chemotactic CheY regulator and ArcB. However, Kato et al. (2007) suggested that ArcA may have another pathway that exerts this regulation. The role of acetyl phosphate has been eliminated as a possible regulatory mechanism (Liu et al., 2009). Further analysis of other regulatory mechanisms, particularly regarding the dynamics of ArcA interaction with ArcB in phosphorylation and dephosphorylation reactions, is needed.

The modeling of ArcA in interaction with ArcB in this study shows that either the dimeric or tetrameric forms of ArcA, only one monomer binds to the Hpt domain. According to available data on the multimerization of ArcA, the complex formation between phosphorylated and unphosphorylated ArcA is 1:1 (Jeon et al., 2001). Therefore, the predictions shown here suggest that in the model of ArcA activation where only one ArcA monomer is needed to be phosphorylated, the other ArcA monomers interact weakly with ArcB awaiting to form an active dimer and then interact with a second activated ArcA dimer to form the DNA-binding ArcA tetramer (Jeon et al., 2001).

Signal sensing

The mechanistic role of Cys180 has been experimentally determined; this residue is needed to form a disulfide bridge that usually is in the range of ~2.4–4.0 Å to form (Chaney & Steinrauf, 1974). Also, the environment near the cysteine residue is involved in disulfide bond formation (Bhattacharyya, Pal & Chakrabarti, 2004). Here, the presence of Tyr170 suggests that the electrons in the aromatic π-system avoid the lone sulfur pair of electrons, modifying the plane at which the disulfide is located (shown here in Figs. 4B and 4C) (Bhattacharyya, Pal & Chakrabarti, 2004). In type 1 kinases, Tyr170 is followed by either Phe169 or Tyr169 (Georgellis et al., 2001); both residues, in the predicted structures, are facing in the opposite direction than Tyr170, suggesting that the presence of aromatic residues hampers the formation of Cys180-Cys241 disulfide bonds and orients both subunits to each other. Also, the location of Phe177 may play a role in the redox state of Cys180 due to the orientation towards this residue. The experimental assessment of the two aromatic amino acids near Cys180 may shed light on the fine-tuning of the regulation of ArcB.

The two types of ArcB kinases found to date suggest that in type 2 kinases, the sensing of the signal may require the interaction with another protein rather than by disulfide bond formation or the involvement of other cysteine residues not located in the linker between the transmembrane domains and the catalytic domains. Further research on organisms bearing type 2 kinases will illuminate this subject.

Overall, in this work, we addressed the following. First, AlphaFold2 models are accurate in two regions that have been experimentally determined for the sensor kinase. Second, the multimer algorithm of AlphaFold2 provided a model consistent with the available experimental evidence on the phosphorelay mechanism of ArcB. Third, CB-Dock analysis suggests that two cavities can be involved in the binding of anaerobic metabolites. The binding depends on the ligands’ cavity size and physicochemical properties. Finally, there is still much work to be done regarding the experimental determination of the structure of both type 1 and 2 ArcB kinases, at least the cytosolic portion of the kinase, to assess the oxidation state and the environment surrounding the cysteine residues.

Conclusions

ArcB/A TCS has been the center for many studies regarding its function and role in cell physiology. Numerous research has been conducted to date addressing the fundamental aspects of signal perception and target genes. In this work, we provide insight on the structural features of the ArcB sensor kinase and ArcA response regulator that are consistent with the available experimental evidence. Protein models used in this work are compatible with the current biochemical and genetic data. However, in this work, we provide a hypothesis that D-lactate and perhaps other anaerobic metabolites may bind to the PAS and catalytic domains, suggesting that experimental approaches to elucidate which two binding-sites are the true regulatory pocket of ArcB. Although models are providing insight into protein function and characterization, crystallographic data is urgently needed to clarify further the mechanistic details of ArcB anaerobic metabolites binding and further characterize the phosphotransfer mechanism to ArcA response regulator.

Supplemental Information

Supplemental Information 1 Supplementary Figures.

Click here for additional data file.

Supplemental Information 2 All protein models used in this work.

Click here for additional data file.

Additional Information and Declarations

Competing Interests

Author Contributions

Data Availability

Bernardo Franco and Rodolfo García-Contreras are Academic Editors for PeerJ. The other authors declare that they have no competing interests.

Felipe Padilla-Vaca conceived and designed the experiments, analyzed the data, prepared figures and/or tables, authored or reviewed drafts of the article, and approved the final draft.

Javier de la Mora conceived and designed the experiments, performed the experiments, analyzed the data, prepared figures and/or tables, authored or reviewed drafts of the article, and approved the final draft.

Rodolfo García-Contreras conceived and designed the experiments, performed the experiments, analyzed the data, prepared figures and/or tables, authored or reviewed drafts of the article, and approved the final draft.

Jorge Humberto Ramírez-Prado performed the experiments, analyzed the data, prepared figures and/or tables, authored or reviewed drafts of the article, and approved the final draft.

Marcos Vicente-Gómez performed the experiments, prepared figures and/or tables, and approved the final draft.

Francisco Vargas-Gasca performed the experiments, prepared figures and/or tables, and approved the final draft.

Fernando Anaya-Velázquez performed the experiments, prepared figures and/or tables, and approved the final draft.

Itzel Páramo-Pérez performed the experiments, prepared figures and/or tables, and approved the final draft.

Ángeles Rangel-Serrano performed the experiments, prepared figures and/or tables, and approved the final draft.

Patricia Cuéllar-Mata performed the experiments, prepared figures and/or tables, and approved the final draft.

Naurú Idalia Vargas-Maya conceived and designed the experiments, analyzed the data, prepared figures and/or tables, authored or reviewed drafts of the article, and approved the final draft.

Bernardo Franco conceived and designed the experiments, analyzed the data, prepared figures and/or tables, authored or reviewed drafts of the article, and approved the final draft.

The following information was supplied regarding data availability:

The protein models are available in the Supplemental Files.

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
