# Peer review of "Theoretical study of ArcB and its dimerization, interaction with anaerobic metabolites, and activation of ArcA"

_PeerJ, doi:10.7717/peerj.16309_

## Round 0.1 · original submission · Major Revisions

Dear Dr. Padilla-Vaca and colleagues:

Thanks for submitting your manuscript to PeerJ. I have now received three independent reviews of your work, and as you will see, one reviewer recommended rejection, while another suggested a major revision. I am affording you the option of revising your manuscript according to all three reviews but understand that your resubmission may be sent to at least one new reviewer for a fresh assessment (unless the reviewer recommending rejection is willing to re-review).

The methods should be clear, concise and repeatable. Please ensure this, and make sure all relevant information and references are provided. Also, elaborate on the discussion of your findings, placing them within a broad and inclusive body of work by the field (relevance to two-component systems and ArcA/B structural biology).

Please fix all of the identified grammatical issues.

Please also clearly identify your research question.

Please note that reviewer 3 has included a marked-up version of your manuscript.

Therefore, I am recommending that you revise your manuscript, accordingly, taking into account all of the issues raised by the reviewers.

I look forward to seeing your revision, and thanks again for submitting your work to PeerJ.

Good luck with your revision,

-joe

Reviewer 1 ·

Basic reporting

Review for: Theoretical study of ArcB and its dimerization, interaction with anaerobic metabolites and activation of ArcA.

Summary:
Authors present the in silico structural analysis of ArcB by using AlphaFold2 model. They showed that results models are consistent with extensive and available data of genetic and biochemical studies of the Arc system. Particularly, they detected putative binding cavities to anaerobic metabolites. However, there is no experimental data provided. They also encourage testing these predictions experimentally.

General comment:

The relevance of the article is not enough for publication in its current state.

The main results of this study are based on the analysis of ArcB by AlphaFold2. They justified this analysis saying that “limited structural data (for ArcB) is available.” However, there is available crystal structure of the domains of the anaerobic sensor kinase ArcB and by searching in AlphaFold database with the accession AF-P0AEC3-F1 (Fig. 1) is possible to get the AlphaFold structure prediction. This fact makes it difficult to determine the real research question of this article. This makes this article look like a review about how the experimental data match with the structural model of this protein.

The most interesting result of this study is about the two putative regulatory cavities for binding anaerobic metabolites in ArcB, however the lack of experimental evidence leads this hypothesis unsupported, thus a rigorous examination of ArcB concerning this subject should be provided to publish this article.


In general, AlphaFold2 tool allows scientists to predict accurate structures for the large majority of proteins. Since the structure of ArcB has been well documented, the analysis of the result model shown here seems weak for a publication. AlphaFold2 is a powerful tool that now can be massively used, but it must be used to determine more complex and interesting biological questions implying experimental approaches to test the hypothesis based on these predicted models.

Experimental design

no comment

Validity of the findings

no comment

·

Basic reporting

1. Literature references are provided appropriately
2. Article structure, figures, tables and raw data are appropriate
3. There are some sentences where language refinement is required, mentioned in the "Additional Comments" section

Experimental design

1. Research is within the scope of the journal
2. Questions and objectives are clearly defined
3. Investigations and data are technically relavent.
4. Methods are sufficient

Validity of the findings

The findings are significant to understand ArcA/ArcB kinase mechanisms and their functions.

Additional comments

1. Lines 81-82, the sentence is incomplete. (Transport, catabolism and more…), it should be concluded appropriately.
2. Line 172, it was mentioned as 21 to 78 as TM helix, however some other reports (eg. PNAS, 2004, 101, 36, 13318) mentioned that the helix is from 22 to 77. Justify
3. Lines 107-109, the sentence should be re-written as this is grammatically incorrect. “These models have enabled to develop of new hypotheses…”
4. Lines 125-126, the sentence is confusing. “Both with the monomeric or dimeric form of ArcB and the ArcA dimer alone”
5. Line 193, and Figure 1 panel C, PAS domain alignment with the experimental structure doesn’t match. Explanation with RMSD and TM are not sufficient, should be justified.
6. Along with Lactate binding, the differences observed in pyruvate and succinate should also be discussed somewhere around lines 340 to 391.
7. Details about the cavity-detection blind docking algorithm should be provided.
8. Lines 615-616 should be re-written. “…a potential cavity that could potentially disrupt…”
9. Line 495, it should be Cys180 not Cys180.
10. Lines 483-489, the phosphotransfer mechanism with ArcA is a bit confusing, should provide more details and data suggesting the mechanism.

Reviewer 3 ·

Basic reporting

The study is focused on studying the ArcA and ArcB system using computational methods and obtaining molecular-level insights.

Experimental design

Methods need to be compiled better; it is very hard to follow in the current state. Maybe including a flowchart highlighting the methods and objectives to use them would make it easier to follow.

Validity of the findings

Most of the findings have been reported based on the computational results.The discussion section needs to be more concise, and the conclusion is missing.

Annotated reviews are not available for download in order to protect the identity of reviewers who chose to remain anonymous.

---

## Round 0.2 · Minor Revisions

Dear Dr. Padilla-Vaca and colleagues:

Thanks for revising your manuscript. The reviewers are very satisfied with your revision (as am I). Great! However, there are a few issues to entertain. Please address these ASAP so we may move towards acceptance of your work.

Best,

-joe

·

Basic reporting

Authors have addressed all the comments appropriately. I have no further comments.

Experimental design

Authors have addressed all the comments appropriately. I have no further comments.

Validity of the findings

Authors have addressed all the comments appropriately. I have no further comments.

Additional comments

Authors have addressed all the comments appropriately. I have no further comments.

Reviewer 3 ·

Basic reporting

The manuscript looks much better now,and authors have made the changes as requested.There are few minor comments :

Line 122 : "provided here" Where it has been reported is not clear?

Line 183 : "Leucine zipper analysis"

Line 498: "Also, the cavity size and Vina score for succinate and pyruvate is surprisingly like D and L-lactate binding sites". ..Not sure what you mean by this.

Experimental design

The study looks concrete with all explanations provided.

Validity of the findings

The findings are well-elaborated and explained.

---

## Round 0.3 · accepted · Accept

Dear Dr. Padilla-Vaca and colleagues:

Thanks for revising your manuscript based on the concerns raised by the reviewers. I now believe that your manuscript is suitable for publication. Congratulations! I look forward to seeing this work in print, and I anticipate it being an important resource for groups studying ArcA/B function. Thanks again for choosing PeerJ to publish such important work.

Best,

-joe